# Spirituality as a Therapeutic Approach for Severe Mental Illness: Insights from Neural Networks

Henderikus Knegtering [1,2,3,4,*], Richard Bruggeman [2,3,4] and Symen Kornelis Spoelstra [5,6]

1. Lentis Research, Lentis Psychiatric Institute, 9725 AG Groningen, The Netherlands
2. University Medical Center Groningen, University of Groningen, 9713 GZ Groningen, The Netherlands; r.bruggeman@umcg.nl
3. University Center of Psychiatry, University of Groningen, 9713 GZ Groningen, The Netherlands
4. Rob Giel Research Center, 9700 RB Groningen, The Netherlands
5. Addiction Care North Netherlands, 9728 JR Groningen, The Netherlands; k.spoelstra@vnn.nl
6. NHL Stenden University of Applied Sciences, 8917 DD Leeuwarden, The Netherlands
* Correspondence: rikusknegtering@gmail.com

**Abstract:** This article explores the link between spirituality/religiosity and mental health from a clinical and neuroscience perspective, taking into account the advancements in neuroimaging. Specifically, it examines how spirituality influences the treatment of mental illness, emphasizing the importance of neuronal networks in cognitive and emotional processes, with a focus on the default mode network (DMN) of the brain. The discussion explores the role of spirituality/religiosity in managing mental disorders and how alterations in the DMN may provide insight into the impact of spirituality/religiosity on mental health. By also discussing spiritual and non-spiritual meditation, as well as spiritual experiences facilitated by the use of psychedelics in psychiatry and the associated brain networks, we aim to elaborate on the importance and limitations of spirituality within psychiatry.

**Keywords:** spirituality; religion; psychiatry; neuroscience; brain; treatment





## 1. Introduction

The intersection of spirituality, mental illness, and neuroscience is a complex and evolving field of research. There is an increasing emphasis on the significance of spirituality in mental health, as evidenced by studies exploring its connections, potential advantages, and therapeutic implications (Koenig 2012; Zimmer et al. 2016; Mosqueiro et al. 2023; van Nieuw Amerongen et al. 2024). Studies are increasingly examining the potential connections between spiritual practices, meditation, brain function, and mental health (van Elk and Aleman 2017). Researchers aim to understand how spiritual beliefs, practices, and experiences may influence mental health outcomes, and they investigate the underlying neurobiological mechanisms that may explain these connections (Rosmarin et al. 2022).

Currently, there is no generally accepted definition of spirituality, and its definitions have evolved over time (McCarroll et al. 2005). The progress of research in this field has been hindered by the absence of a unanimous agreement on the definition of spirituality and a lack of financial support (Pesut et al. 2008; Rosmarin et al. 2022). Traditionally associated with religious processes connecting humans to a higher power, spirituality now extends to encompass an individual's connection to a sense of purpose, meaning, and transcendence. This may often involve beliefs in a connection to nature or a sense of belonging to something greater than oneself (McMahan 2008; Morgan 2010). Spirituality is often conceptualized within the human spirit or soul rather than within material or physical things (Waaijman 2002; Sheldrake 2007). Thus, one can experience spirituality by looking at the larger picture of life and asking existential questions about life. It may also refer to a subjective experience of a sacred dimension (Saucier and Skrzypinska 2006)

and the "deepest values and meanings by which people live" (Waaijman 2002; Sheldrake 2007). In the context of this article, spirituality is broadly characterized as an individual's subjective encounters with a "higher power" and/or their sense of connection with the transcendent (Rosmarin et al. 2022).

Potential associations between spirituality, religion, and mental health or well-being have been increasingly studied over the past decades (van Elk and Aleman 2017; Rim et al. 2019; Jetan et al. 2023). Numerous studies have supported various positive outcomes of mental health and overall well-being related to spirituality and/or religiosity (Rosmarin et al. 2022). Religious beliefs and commitments, which are aspects of spirituality, have been shown to help individuals cope with stressful life events, reduce death anxiety, enhance life satisfaction, improve adjustment, and decrease the likelihood of developing depression and anxiety (Rim et al. 2019). In addition, spirituality is also associated with a reduced number of suicides and less severe alcohol or drug use (Rosmarin et al. 2022; Griffiths et al. 2023; Beraldo et al. 2019; Čepulienė and Skruibis 2022). The bidirectional influence of spirituality and religion on mental health highlights the need for further research. However, aside from positive effects, spirituality can also have negative effects, such as influencing the presentation of psychotic and obsessive symptoms (Rosmarin et al. 2022). In addition, there are also aspects of spirituality and religion, such as alienation, struggle, and a sense of sin, that can be negatively associated with mental health (Gallardo-Vergara et al. 2022). Furthermore, psychiatric symptoms with religious or spiritual content (delusions or hallucinations) may occur (Hoenders and Braam 2020). Interestingly, individuals with a significant connection to religious and spiritual practices may experience positive treatment outcomes when interventions specifically address the dimensions of religion and spirituality (Bouwhuis-Van Keulen et al. 2024). In 2016, the World Psychiatry Association urged the integration of spirituality and religion within psychiatry for a more holistic mental health care approach (Moreira-Almeida et al. 2016).

This article investigates the relationship between spirituality/religiosity and mental health from a clinical and neuroscience perspective, considering advancements in neuroimaging techniques. Specifically, it examines how spirituality influences the treatment of mental illness, emphasizing the significance of neuronal networks in cognitive and emotional processes, with a specific focus on the default mode network (DMN) (Broyd et al. 2008). The discussion delves into the role of spirituality/religiosity in managing mental disorders and how alterations in the DMN may offer insight into the impact of spirituality/religiosity on mental health. Additionally, by discussing both spiritual and non-spiritual meditation, as well as spiritual experiences facilitated by the use of psychedelics in psychiatry and their associated brain networks, we aim to elaborate on the importance of spirituality within psychiatry.

## 2. Psychological Theories on Spirituality

Influential figures in the field of mental health, like Sigmund Freud, considered spirituality as a neurotic development, while Carl Jung, Viktor Frankl, and William James held more positive views on spiritual experiences (Rosmarin et al. 2022). Both Freud and Skinner perceived facets of human behavior like initiative and social behavior as predominantly unconscious and biologically driven and essential to procreate and survive as individuals and species (Sulloway 1992). From their perspective, primarily genetic-based behavior patterns are shaped through parenting, social interactions, and society (Sulloway 1992). According to Freud, psychological challenges may arise from conflicts between biological-originated drives and behaviors with demands and restrictions from the social environment and society (Solms 2018). As these demands and restrictions may also be rooted in a person's religious or spiritual culture, these aspects are not necessarily helpful or can even contribute to the emergence of psychiatric problems (Rosmarin et al. 2022). In contrast, Skinner tends to explain the development and reinforcement of behavior as a primary reaction of biologically determined behaviors that can be reinforced in reaction to interaction with the environment, including other individuals. The theories of Freud and

Skinner are often considered to be based too much on deterministic biological principles, reflecting a Western-centric perspective on biology and psychopathology. In modern psychiatry, social, cultural, and spiritual dimensions of human well-being are thought to be under-represented in the work of Freud, Skinner, and their followers (Rose 1996). Their approach is contemplating the more positive aspects of human development in a cultural and spiritual framework (Rose 1996; Ryan and Deci 2000).

### 3. The Loss of Purpose, Meaning, and Initiative

When individuals face severe unsolvable physical or psychological challenges, they may experience a reduction in their ability to engage with the world, leading to a loss of interest in activities and social withdrawal (Aleman and Lanctôt 2021). Moreover, spirituality and religious activities may be severely diminished when symptoms such as apathy, demoralization, or so-called negative symptoms in psychosis dominate the individual's reaction or psychiatric conditions (van Elk and Aleman 2017). This decline in initiative can then create a feeling of uselessness, contributing to sadness and apathy, ultimately leading to a loss of purpose or meaning. Conditions such as severe depression, trauma, psychosis, and dementia can exacerbate this lack of initiative and apathy, resulting in significant social and emotional challenges (Kos et al. 2016; Aleman and Lanctôt 2021). As these issues impact social functioning (including spiritual/religious activities) and overall well-being, addressing them is crucial for both patients and caregivers.

The interaction between apathy, depression, anxiety, trauma, psychosis, and spirituality hints at possible connections between these factors, potentially offering insights into treatment in mental health. The engagement of specific neuronal networks, such as the default mode network, may be crucial in understanding how spiritual practices and beliefs influence the brain and could potentially serve as targets for the treatment of psychiatric symptoms including anxiety, depression, apathy, and addiction.

### 4. Brain Networks in Severe Mental Illness

Brain functions can be studied by analyzing how (large) groups of brain cells function together by grouping scattered brain areas to perform certain tasks. Those groups of cells and brain areas functioning as closely linked to each other are often called (functional) brain networks. As we will briefly review, several brain networks have been implicated in major psychiatric problems (V. Menon 2011; B. Menon 2019; Broyd et al. 2008). Research on brain networks is continuously evolving and has the potential to transform our understanding of the brain, including our understanding of psychiatric disorders. The functional connections within and between networks of the brain may help us understand human behavior and how environmental factors interact with brain functioning. In a similar vein, the identification of neural networks involved in psychiatric problems is hoped to contribute to the development of more targeted and effective treatments. Furthermore, studying the connectivity of the brain in reaction to treatment options, including psychotherapy, pharmacotherapy, mindfulness-based therapy, and spirituality may offer insights into their putative mechanisms of action. In this regard, we focus on the three most studied networks of the brain related to psychiatric problems, with a focus on the default mode network.

#### 4.1. Default Mode Network (DMN)

The identification of the default mode network in the brain (DMN) in 1998 revolutionized our understanding of the human brain. The DMN consists of synchronized activated groups of neurons in the brain that are active during self-reference, social cognition, episodic and autobiographical memory, language and semantic memory, and mind wandering (Christoff et al. 2009). The DMN is thought to integrate memory, language, and semantic representations in order to create an internal picture reflecting our individual experiences. Furthermore, the DMN is involved in shaping our sense of self, influencing how we perceive ourselves and interact with others; hence, it forms a vital component of human consciousness (V. Menon 2023). The DMN is particularly active when we are not

focused on external stimuli, facilitating self-referential processing, such as thinking about our thoughts and feelings. Disruptions within the DMN have been linked to a variety of psychiatric disorders, including depression, anxiety, addiction, psychosis, and autism spectrum disorder (ASD) (Broyd et al. 2008; V. Menon 2011, 2023; Zhang and Volkow 2019).

### 4.2. Salience Network (SN)

The salience network (SN) is thought to be involved in detecting and responding to salient (important) stimuli, both internal and external. It helps us to focus our attention on important information while filtering out distractions. Disruptions within the SN have been linked to a variety of psychiatric disorders, including schizophrenia, depression, and anxiety (Seeley 2019).

### 4.3. Frontoparietal Network (FPN)

The frontoparietal network (FPN), also known as the Central Executive Network (CEN), is involved in executive functions, such as planning, problem-solving, and decision-making. It also helps us to control our thoughts and emotions. Disruptions within the FPN have been linked to a variety of psychiatric disorders, including schizophrenia, attention deficit hyperactivity disorder (ADHD), and obsessive compulsive disorder (Marek and Dosenbach 2018).

In addition to these three networks, other brain structures have also been implicated in different tasks related to brain function and psychiatric disorders. For example, the amygdala is crucial in processing emotions, while the hippocampus is involved in memory formation. However, these networks are beyond the scope of this article.

## 5. Switching between Networks in the Brain

Pivotal to both physical and mental health, an organism has to be able to adapt quickly to the ever-changing internal and external demands and threats encountered in life. Regarding brain networks, mental health is partly reflected by the ability to switch fast between various brain networks, including between the three above-mentioned networks: the salience network for prioritization, the frontal–parietal network for executing actions, and the default mode network for integration and evaluation. Many severe mental disorders are characterized by reduced flexibility to switch between mental states and the auxiliary brain networks. As an illustration, patients with chronic, therapy-resistant depression often tend to show alterations in the resting state functional connectivity of the DMN, suggesting a degree of inflexibility (Grehl et al. 2023). In contrast, people improving from severe mental disorders may regain mental flexibility, reflected by increased flexibility between active neural networks.

### 5.1. Exploring the Role of Brain Networks in Spirituality, Psychedelics, and Mindfulness as Therapeutic Interventions

Exploring the convergence of therapeutic interventions, spirituality, and their neural reflections may unveil a fascinating intersection in mental well-being. Examining the interplay between various therapeutic approaches and personal spirituality within the complex networks of the brain may offer insights for nuanced and effective strategies that address both the psychological and neurological aspects of overall well-being. The possible influence of spirituality, psychedelics, and mindfulness on brain networks are discussed below.

#### 5.1.1. Spirituality/Religiosity

Aggarwal et al. (2023) conducted a systematic review examining the role of religiosity and spiritual involvement (formal and informal) in the prevention and management of depression and anxiety in young people aged 10 to 24 years. They found that religiosity and spirituality may be effective in the prevention and management of mental illness, especially depression and anxiety disorders. Interventions incorporating religious and

spiritual practices for depression and anxiety in young people were generally effective, although the overall study quality was typically low and the heterogeneity in study designs precluded meta-analyses. Individuals with lived experience described spirituality and religious engagement as central to their way of life and greatly valued the "feeling of being watched over" during difficult times. While religiosity and spirituality have been associated with positive mental health outcomes in adults, there is also evidence that suggests that aspects of religiosity may be associated with poor mental health outcomes, partly influenced by personal histories (Aggarwal et al. 2023).

Despite the abundant literature on religiosity/spirituality and health, few studies have explored their clinical impact on anxiety. A meta-analysis by Gonçalves et al. (2015) examined the effectiveness of religious/spiritual interventions on anxiety and depression, as shown in randomized clinical trials. Their analysis identified 23 relevant studies, indicating significant benefits of these interventions in reducing anxiety, stress, alcoholism, and depression. These findings emphasize the potential of religious/spiritual interventions in mental health care, highlighting the need for further research and intervention standardization.

Various human conditions, such as attention deficit hyperactivity disorder (ADHD), depression, anxiety, and addiction, have been associated with increased activity of the DMN (Roy et al. 2021; Mowinckel et al. 2017). A study by Svob et al. (2016) found that individuals at high risk for depression exhibit increased connectivity within the DMN. Interestingly, a belief in the importance of religion/spirituality is associated with lower DMN connectivity in this group. Adopting religion/spirituality as personally important may have adaptive effects on the DMN and could be linked to resilience in individuals at risk for depression. Importantly, this protective effect of religious/spiritual importance seems to be specific to DMN connectivity and is not observed in the Central Executive Network (=frontal–parietal network) (CEN) (Broyd et al. 2008). This finding has implications for depression therapies, especially those involving meditation, which often rely on DMN-CEN connectivity (Svob et al. 2016).

A characteristic of depression is the tendency to ruminate or mentally repeatedly dwell on the negative aspects of one's life. This type of maladaptive rumination is thought to be a tendency of depressed people, but it may also perpetuate depression by focusing primarily on the negative, whilst ignoring the positive aspects in life. Studies on depression have found increased activity and connectivity in the DMN during maladaptive rumination, which potentially might exacerbate depressive symptoms (Svob et al. 2016; Grehl et al. 2023; Healy 2021).

Yan et al. (2019) reported reduced resting-state functional connectivity within the DMN in patients with major depressive disorder, while Carhart-Harris et al. (2017) reported an increase in functional connectivity following treatment with psilocybin. These studies suggest that disrupted functional integration of the DMN may play a role in psychiatric disorders, and effective treatments could lead to the restoration of this network.

A review of Rosmarin et al. (2022), including 18 studies, highlights the protective role of religion against various mental health problems in adults. The reviewed studies indicate that religion is associated with a lower risk of depression, especially among those with familial predisposition. Moreover, religiosity appears to be linked to various neurobiological factors, including decreased DMN activity (Svob et al. 2016; Galanter et al. 2017). The literature indicates that neural features may serve as mediators in the relationship between familial depression and the onset of depression (Svob et al. 2016), and religion might play a role in addressing these vulnerabilities. Specifically, research by Galanter et al. (2017) showed that participation in an experimental prayer group correlated with decreased alcohol cravings when exposed to alcohol-related stimuli. The associated increase in brain activation in specific regions suggested that prayer might contribute to reduced alcohol cravings and enhanced attention and control processes (Galanter et al. 2017). However, the review acknowledges limitations in the available data concerning anxiety and psychosis. Only three publications were included, with two studies on anxiety reporting null findings and only one study exploring non-clinical psychosis. No studies have yet examined clinical

psychotic symptoms in relation to spirituality/religion and neurobiological correlates. Thus, conclusive statements on the interplay between spirituality, neurobiology, anxiety, and psychosis are challenging due to limited data, stressing the importance of further research. Altogether, the accumulating evidence suggests a connection between spirituality/religion and positive mental health outcomes, particularly a reduction or change in the DMN connectivity (Svob et al. 2016; Yan et al. 2019).

A commonly used intervention for alcohol use disorders is Alcoholics Anonymous (AA), which can be considered a spirituality-based intervention contributing to recovery (Beraldo et al. 2019). Spirituality is a fundamental aspect of AA, integrated into most of the 12 steps and 12 traditions of AA groups (Kelly 2017). This suggests that spirituality and religiosity can positively influence treatment outcomes in alcohol dependence (Mcclintock et al. 2019). Addiction influences the activation of the DMN (Zhang and Volkow 2019). Spiritual interventions have been shown to positively impact treatment outcomes for patients with substance use disorders (SUDs) (Beraldo et al. 2019).

Some studies suggest that the DMN is overactive in individuals with psychotic symptoms, especially schizophrenia (Guo et al. 2017). One hypothesis is that people with schizophrenia have difficulties in shifting their thought patterns away from internally focused thinking, a crucial characteristic of the default mode network. Additionally, overactivity in the DMN might reflect a difficulty in distinguishing between thoughts and sensory perceptions, which could contribute to hallucinatory experiences (van Ommen et al. 2023; Marino et al. 2022). Although psychotic symptoms often include religious and spiritual experiences, the possible role of spirituality/religion in coping with psychotic symptoms is under-studied (Rosmarin et al. 2022).

### 5.1.2. Psychedelic Experiences

The treatment of mental disorders with psychedelics has regained new attention (Gattuso et al. 2023). Many psychedelics have the potential to trigger mystical experiences, commonly characterized as spiritual and transcendent. Following their usage, individuals often express experiences of spiritual development and a heightened sense of connection with nature, fellow humans, and the universe (Griffiths et al. 2023). Indeed, specific religious practices call upon these substances to obtain spiritual insights or to engage in communication with the divine or a higher power and result in spiritual breakthroughs (Yaden and Griffiths 2021; Nayak et al. 2023; Olson 2021).

In many cultures, psychedelics like psilocybin, ayahuasca, and mescaline are essential elements of traditional medicine to assist people in their personal development, religious and spiritual experiences, and mental health. Nowadays, psychedelic substances are increasingly studied in Western medicine, which is trying to improve mental health and well-being in psychiatric, addiction, and physical medicine, including interventions for end-of-life anxiety (Geyer 2023; Moreton et al. 2023). Studies suggest that spiritual experiences during treatment with psychedelics correlate with improved well-being and mental health and a decrease in end-of-life anxiety (Yaden and Griffiths 2021; Nayak et al. 2023; Olson 2021). A crucial yet unanswered question is whether the potential beneficial effects of psychedelics are to be ascribed to their ability to promote spiritual and religious experiences or to a more biological mechanism (Geyer 2023; Barksdale et al. 2024).

Most studies of psychedelics have used functional magnetic resonance imaging (fMRI) with network connections assessed using functional connectivity (FC). FC measures correlations between Blood-Oxygen-Level-Dependent (BOLD) signal fluctuations as a proxy for brain activity. As stated earlier, alterations in DMN connectivity may underlie the excessive internal focus and rumination characteristic of depression. More specifically, DMN hyper-connectivity has been reported in patients with major depression and correlates positively with rumination. Recent imaging data indicate that psychedelics generally reduce connectivity within the DMN, enlarging the involvement of other brain areas involved in planning and social functioning (prefrontal cortex areas), and are correlated with improved mood and social functioning (Husain et al. 2023).

Psychedelics, known for causing vivid hallucinations and profound psychological effects, are well studied regarding their impact on the DMN in the brain. Studies on LSD, psilocybin, and ayahuasca consistently show disruptions in DMN connectivity and increased functional connectivity between resting-state networks. Various mechanisms have been proposed to explain the cognitive mechanisms of psychedelics, and in one model, DMN modulation is a central axiom. Although the DMN is consistently implicated in psychedelic studies, it remains unclear how central the DMN is to the therapeutic potential of classical psychedelic agents (Gattuso et al. 2023).

In recent years, there has been growing evidence that classic psychedelics may also be effective in the treatment of addiction, leading to growing interest in their potential as pharmacological treatments in addiction (Mertens and Preller 2021; Johnson 2022; Zafar et al. 2023).

Of interest, findings in addiction research with fMRI suggest that a distributed functional neural network in the brain is associated with spiritual experiences and provides a foundation for investigating brain mechanisms underlying the role of spirituality in recovery from behavioral addictions (Mcclintock et al. 2019).

Overall, research indicates that spiritual experiences during the use of psychedelics predict positive outcomes, such as improved mood, reduced anxiety, and belief shifts, which may be associated with reducing cravings and increasing self-efficacy, supporting the reduction in substance abuse (Yaden and Griffiths 2021; Nayak et al. 2023; Olson 2021). Factors like individual pharmacological actions, context, motivations, and therapy integration impacting subjective experiences are all thought to be important. As stated earlier, researchers conclude that psychedelic (spiritual, mystical, and religious) experiences predict therapeutic effects in managing addiction, but others tend to regard these experiences just as non-essential side effects (Yaden and Griffiths 2021; Olson 2021; Barksdale et al. 2024). Both the spiritual and the pharmacological explanations of the effects of psychedelics on the brain may help us understand their mechanisms of action (Geyer 2023; Barksdale et al. 2024).

### 5.1.3. Meditation and Mindfulness

Neuroscientific studies utilizing techniques like functional magnetic resonance imaging (fMRI) have identified specific brain regions associated with spiritual experiences, such as meditation and prayer. The mind–body connection is crucial in understanding how practices like meditation and mindfulness, often rooted in spirituality, may positively impact brain structure and function, influencing emotional regulation and stress response (van Elk and Aleman 2017). Individual experiences of spirituality and mental health can vary widely, and cultural and personal factors play a significant role in shaping these connections (van Elk and Aleman 2017).

The results of a study by Garrison et al. suggest that meditation is associated with reduced activity in the DMN compared to an active task in meditators as opposed to controls. Regions of the DMN involved include the posterior cingulate/precuneus and anterior cingulate cortex (Garrison et al. 2015). The concept of neuroplasticity highlights the brain's ability to reorganize itself, and spiritual practices may contribute to such changes in neural pathways (Mohandas 2008). Spiritual practices, including prayer and meditation, have been linked to reduced levels of the stress hormone cortisol, offering potential benefits for mental health by targeting stress reduction (Sobhani et al. 2022). Additionally, spirituality has been associated with enhanced resilience in the face of life challenges, and ongoing research aims to understand the neural mechanisms underlying this resilience (Rim et al. 2019).

There are evident similarities and differences between various forms of mindfulness meditation compared to different spiritual practices (Saucier and Skrzypinska 2006). A review by Barnby and colleagues suggests that spiritual experience and non-spiritual mindfulness may have distinct representations within the brain (Barnby et al. 2015). Thus, a relative increase in prefrontal activation was associated with mindfulness, which was

related to decreased anxiety and improved well-being. Likewise, a relative decrease in the activation of the inferior parietal cortex was correlated to spiritual belief, whether within the context of meditation or not. Barnby and co-authors suggest that the findings from neuroscience research may reveal an important difference between mindfulness meditation and spiritual practices, i.e., their focus on "self" versus "other", as reflected by their differential representations in the brain. On the other hand, both practices may relatively activate other networks in the brain while decreasing the activity in the DMN, suggesting a common pathway.

## 6. Discussion

Numerous qualitative and quantitative studies have highlighted the importance of the terms "meaning" and "spirituality" in individuals' lives, indicating the desire of mental health professionals to address these topics, including in severe psychiatric disorders and crisis situations (Oxhandler et al. 2018).

Here, we have explored the relation between spirituality and psychiatric disorders and the putative underlying alterations in brain networks, with a specific focus on the DMN. Several psychiatric conditions, including depression, anxiety, and addiction, are associated with increased activity in the DMN (Broyd et al. 2008). At the same time, there is evidence that spirituality/religious activities, treatment with psychedelics (often associated with spiritual experiences), and mindfulness (with and without spiritual experiences) can influence brain network activity, especially by decreasing activity in the DMN, regaining functional flexibility between networks and increasing the relative activities of the FPN and/or SN. From a clinical point of view, these changes coincide with less attention to inner experiences and increased openness to external experiences, facilitating adaptation to the outside world.

Many studies suggest that spirituality has positive effects in alleviating psychiatric symptoms and helps individuals cope with psychiatric problems. However, spiritual experiences may not be helpful in all cases, depending on earlier life experiences with religion and spirituality (Gallardo-Vergara et al. 2022; Aggarwal et al. 2023; Rosmarin et al. 2022).

Based on the reviewed literature, we conceptualize that spiritual/religious activities can assist people in reconnecting with the external world, particularly those who identify themselves within religious and spiritual contexts (Bouwhuis-Van Keulen et al. 2024). Especially when psychiatric symptoms may lead to dysfunctional religious and spiritual beliefs and experiences, often accompanied by social problems, religious and spiritual guidance may help reattach one to the outside world.

On the other hand, both excessive engagement in religiosity and spirituality, as well as a disconnection with beliefs within one's social environment, may generate internalized conflicts or disconnection from the social environment and may potentially contribute to psychiatric problems (Gallardo-Vergara et al. 2022).

It is tempting to perceive the reflection of dysfunctional social and psychological networks in brain networks. The engagement of brain networks, including the DMN, is not isolated but depends on factors such as culture, predisposition, upbringing, and other social influences. Spirituality and religiosity can aid individuals with psychiatric problems in transitioning from their inner world (DMN) to the external world (SN). Research focusing on this transition may assist in understanding the underlying neurobiological, psychological, and environmental factors such as the culture and context of psychiatric problems. Religion and spirituality, as well as mindfulness, meditation, and some psychedelics, ideally integrated into a broader treatment approach, may help individuals cope with stress and life's challenges.

Although a significant body of the discussed research supports the psychological benefits of religion and spirituality, de Oliveira Maraldi points out sources of bias in the present research, arguing that despite compelling empirical support for the positive effects of spirituality/religiosity on mental health, there are dangers of response bias in the research on religion, spirituality, and mental health (de Oliveira Maraldi 2020). To add

nuance, we must acknowledge that the current level of research and the commonly used research methods generally do not provide exact insights into the role of religious/spiritual practices, beliefs, and experiences in intervention processes aimed at improving mental health. This is because there are no unique indicators for the effects of these influences, making it difficult to attribute changes in psychobiological markers (such as cortisol levels) or in neural networks solely to these factors.

Another important disclaimer here is that, for those who have no connection to or even dislike religion, or for those who have traumatic experiences linked to religion, spirituality, or members of their faith, spirituality/religious interventions may work against improving their mental health (Gallardo-Vergara et al. 2022; Aggarwal et al. 2023). Future research should focus on exploring how and for whom guidance from a religious and spiritual perspective may be helpful or harmful to decrease psychiatric symptoms and improve coping.

While our research advocates for the continuation of research programs exploring the relationship between spirituality/religiosity, mental illnesses, and neural circuits, we acknowledge its limitations. To create a broader scope on the interplay between psychiatric problems, neuroscience/neuroimaging, and the role of spirituality/religiosity, we did not attempt to do a systematic review of all elements discussed, in an attempt not to get lost in details. For the same reason, several neuroimaging techniques like Electroencephalography (EEG) or Positron Emission Tomography (PET) are not discussed.

Psychiatric symptoms may include many symptom dimensions including anxiety, stress, apathy, depression, addiction, and psychosis. In view of the scope of our article, we could not discuss the influence of spirituality/religiosity interventions on all dimensions separately.

Our article attempts to provide an understanding of the complex connection between these elements, but we must be cautious not to overestimate the role of neural circuits in explaining all psychiatric, religious, or spiritual phenomena. Additionally, the role of neuroimaging in the understanding, diagnosis, and treatment of mental disorders remains a subject of study and debate. We recognize that the current attempt to connect spirituality/religion to mental illnesses and neural circuits is a preliminary effort, and hopefully, it may inspire other researchers to join the discussion. In addition, an important question is whether we can simply demonstrate the effects of spirituality and religion by dissecting how neural circuits respond to meditation or religious rituals, or to what extent these often beneficial effects are related to something more intrinsic to religious belief, and social aspects of mental health, that not simply can be reduced to neural activity.

Nevertheless, it is intriguing that spirituality/religiosity, as well as psychedelics and mindfulness, influence the activity of the DMN, and that in psychiatric disorders, the DMN can exhibit both increased and decreased activation. This suggests, in our opinion, a common pathway of interventions. More insight into brain networks may provide insight into mechanisms and help to optimize approaches, both from a psychiatric and spiritual/religious perspective.

## 7. Conclusions

The interplay between spirituality/religiosity, mental health, and neuroscience highlights the continuously evolving complexity in our comprehension of mental well-being from a holistic perspective. The increasing recognition of spirituality/religiosity in mental health, together with advancements in neuroscience and the investigation of spirituality, opens new pathways for comprehensive therapeutic approaches. As this field progresses, the promising intersections between spirituality/religiosity, psychiatric interventions, and brain function will hopefully provide insightful avenues for enhancing mental health outcomes.

**Author Contributions:** Conceptualization and methodology, H.K. and R.B.; writing—original draft preparation, H.K., S.K.S. and R.B.; writing—review and editing H.K. and S.K.S. All authors have read and agreed to the published version of the manuscript.

**Funding:** This research received no external funding.

**Institutional Review Board Statement:** Not applicable.

**Informed Consent Statement:** Not applicable.

**Data Availability Statement:** Not applicable.

**Conflicts of Interest:** The authors declare no conflict of interest.

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
