# Peer review of "Spirituality as a Therapeutic Approach for Severe Mental Illness: Insights from Neural Networks"

_religions, doi:10.3390/rel15040489_

Round 1

Reviewer 1 Report

Comments and Suggestions for Authors

Spirituality as a Therapeutic Approach for Severe Mental Illness: insights from neural networks

Thank you for the opportunity to review the article

The article is well written, with a logical sequence of content. There is a good discussion of the studies presented.

Despite the inclusion of an extensive list of references, I understand that some important studies were not included and that some divergences in the literature could therefore be important to describe in the findings of this study.

The most important aspect that has a negative impact on this study is that the study is not a systematic review, and the inclusion criteria for the studies presented were not described.

However, I reinforce the importance of the study

Different criteria were used for references. It must be evaluated carefully. Reference 68/69 must be rewritten

Author Response

Thank you very much for your comments. We have adjusted the references.

Although systematic reviews are valuable for synthesizing evidence across a broad range of studies, they may not always be the most suitable approach for every research question or context. In our study, considering the scope of trying to integrate several areas of expertise, we determined that alternative methods would better serve the objectives of our study. We revised the discussion of the article where we elaborated on our choices and possible disadvantages in more detail in the final part of the discussion section.

We added the following text to the discussion:

While our research advocates for the continuation of the research program exploring the relationship between spirituality/religiosity, mental illnesses, and neural circuits, we acknowledge its limitations. To create a broader scope on the interplay between psychiatric problems, neuroscience/neuroimaging, and the role of spirituality/religiosity, we did not attempt to do a systematic review of all elements discussed, in an attempt not to get lost in details. For the same reason several neuroimaging techniques like Electroencephalography (EEG) or Positron Emission Tomography (PET), are not discussed.

Psychiatric symptoms may include many symptom dimensions including, anxiety, stress, apathy, depression, addiction, and psychosis. In view of the scope of our article, we could not discuss the influence of spirituality/religiosity interventions on all dimensions separately. 

Our article attempts to provide an understanding of the complex connection between these elements, but we must be cautious not to overestimate the role of neural circuits in explaining all psychiatric, religious, or spiritual phenomena. Additionally, the role of neuroimaging in the understanding, diagnosis, and treatment of mental disorders remains a subject of study and debate.

We recognize that the current attempt to connect spirituality/religion to mental illnesses and neural circuits is a preliminary effort, and hopefully, it may inspire other researchers to join the discussion. Besides, an important question is whether we can simply demonstrate the effects of spirituality and religion by dissecting how neural circuits respond to meditation or religious rituals, or to what extent these often beneficial effects are related to something more intrinsic to religious belief, and social aspects of mental health, that not simply can be reduced to neural activity.

Reviewer 2 Report

Comments and Suggestions for Authors

The present attempt to connect spirituality/religion (S/R) with mental illness and neural circuitry is interesting and a good attempt, even if not always convincing. I would like to discuss several point with the authors, as a way to engage with their research and to pursue their own program.

The first issue that draws my attention is the attempt to reveal through neuronal networks the dynamics that underpin the possible influence of S/R on mental health and related mental phenomena. The article makes the case for that connection which has been strongly shown in the last years and in many studies. The present article has the merit to provide the freshest literature. However, I am not so convinced that that link is better revealed resorting to the neural activity that could be determined by S/R, and which would govern mental disorders. For instance, the discussion has not being settled regarding to what extent neuroimaging contributes to an accurate diagnosis of mental illness. Apparently it assists in discarding related issues or clarifying involved factors, but less to a diagnostic of many mental disorders.

The authors seem to be less aware about how much research on religion and anxiety has been published in recent years (just looking to PubMed 896 entries under such heading). Our team was showing how religious practice provided meaning during look-down in last pandemics, and so reduced anxiety.

It is possible that reducing anxiety and stress could be assessed controlling for cortisol levels, and this would be already a legitimate path to study effects for instance, of religious prayer or spiritual meditation. But this is not to state that religion reduces stress because it reduces the flow of cortisol. By the same token, that spiritual activity reduces DMN connectivity does not mean that this effect explains the positive side of that practice. The big issue indeed is whether we can explain such healthy benefits indirectly, and so just showing how neural circuits respond to meditation or to religious rituals; or to what extent those healing effect is connected to something more intrinsic in religious faith, and hard to reduce to neuronal activity.

Overall the article makes a good case for pursuing that research program, but in my opinion it could be somewhat limited, as it could explain only a small side of that connection, which probably involves neuronal circuitry, but we can hardly describe it in such terms, as it has been almost impossible to describe religious or spiritual activity in neuronal terms, despite the many attempts in the last 30 years.

The present attempt to link spirituality/religion (S/R) with mental illness and neural circuits is interesting and a good attempt, though not always convincing. I would like to discuss several points with the authors as a way of engaging with their research and pursuing their own programme.

The first issue that catches my attention is the attempt to use neural networks to reveal the dynamics that underlie the possible influence of S/R on mental health and related mental phenomena. The article argues for this link, which has been strongly demonstrated in recent years and in many studies. The present article has the merit of providing the most recent literature. However, I am not so convinced that this link is better demonstrated by relying on the neural activity that could be determined by S/R and that would govern mental disorders. For example, the debate about the contribution of neuroimaging to the accurate diagnosis of mental illness has not been settled. Apparently, it helps to rule out related problems or to clarify the factors involved, but less in diagnosing many mental disorders we know.

The authors seem less aware of how much research on religion and anxiety has been published in recent years (just looking at PubMed 896 entries under this heading). Our team has shown how religious practice provided meaning and reduced anxiety during recent pandemic.

It is possible that the reduction in anxiety and stress could be assessed by controlling for cortisol levels, and this would be a legitimate way to study the effects of, for example, religious prayer or spiritual meditation. But this does not mean that religion reduces stress because it reduces the flow of cortisol. Similarly, the fact that spiritual activity reduces DMN connectivity does not mean that this effect explains the positive side of this practice. Indeed, the big question is whether we can explain such health benefits indirectly, by simply showing how neural circuits respond to meditation or religious rituals, or to what extent these healing effects are related to something more intrinsic to religious belief and difficult to reduce to neural activity.

Overall, the article makes a good case for pursuing this research programme, but in my opinion it could be somewhat limited in that it could only explain a small side of this connection, which probably involves neural circuits, but we can hardly describe it in those terms, because it has been almost impossible to describe religious or spiritual activity in neural terms, despite many attempts over the last 30 years.

My advice is that the article makes a more modest case for that connection, and that the discussion section acknowledges the signaled limits.

Reviewer 3 Report

Comments and Suggestions for Authors

I commend the authors for this valuable review of neuro science evidence and how it may inform treatment approaches in the field of mental health. I found the authors' presentation insightful and helpful. I think the discussion is balanced and the conclusions are cogent. I was first surprised that the authors referred as far back as Freud and Skinner; however, given the tendency of the field of psychiatry to dismiss the spiritual dimension in the past, it is fitting to acknowledge this possible systematic bias that may still influence treatment approaches in the area of mental health.

I'm not prepared to offer much critical feedback. The only stumbling block I detected was in the second paragraph of the introduction where the authors refer to the transcendend dimension as "upper being". I had never come across this word in English though it translates perfectly and literally to what the Germans refer to as "Höheres Wesen". In the context of English publications, I have typically read about "Higher Power" in this context which is a term the authors also use in a different sentence. Maybe stay consistent in terminology?

Author Response

I commend the authors for this valuable review of neuroscience evidence and how it may inform treatment approaches in the field of mental health. I found the authors' presentation insightful and helpful. I think the discussion is balanced and the conclusions are cogent. I was first surprised that the authors referred as far back as Freud and Skinner; however, given the tendency of the field of psychiatry to dismiss the spiritual dimension in the past, it is fitting to acknowledge this possible systematic bias that may still influence treatment approaches in the area of mental health.

Authors

Thank you very much for the time you took to read our article in detail. We also thank you for your comments and compliments.

I'm not prepared to offer much critical feedback. The only stumbling block I detected was in the second paragraph of the introduction where the authors refer to the transcendend dimension as "upper being". I had never come across this word in English though it translates perfectly and literally to what the Germans refer to as "Höheres Wesen". In the context of English publications, I have typically read about "Higher Power" in this context which is a term the authors also use in a different sentence. Maybe stay consistent in terminology?"

Authors

Thank you very much for your additional comments on the term used by us “upper being”. The term “upper being” is indeed not so much used in the English language. We checked it online, in several dictionaries and databases, while “higher power” is used more often. As suggested by reviewer 3, also to be consistent in the text, we replaced “upper being” with “higher power”.

Round 2

Reviewer 2 Report

Comments and Suggestions for Authors

I am quite satisfied with your comments in the Discussion section of your article, even is possibly some note could be added at the begining too, but I leave it to the authors.

Author Response

Dear reviewer,

Thank you again for your time to review our article. In response to your comment and the comments of the editor, we added some text to the discussion as suggested at the end of page 8: 

To add nuance, we must acknowledge that the current level of research and the commonly used research methods generally do not provide exact insights into the role of religious/spiritual practices, beliefs, and experiences in intervention processes aimed at improving mental health. This is because there are no unique indicators for the effects of these influences, making it difficult to attribute changes in psychobiological markers (such as cortisol levels) or in neural networks solely to these factors.